# Squeezing Dynamic Mechanism of High-Viscosity Droplet and its Application for Adhesive Dispensing in Sub-Nanoliter Resolution

**DOI:** 10.3390/mi10110728

**Published:** 2019-10-28

**Authors:** Ping Zhu, Zheng Xu, Xiaoyu Xu, Dazhi Wang, Xiaodong Wang, Ying Yan, Liding Wang

**Affiliations:** 1Key Laboratory for Micro/Nano Technology and System of Liaoning Province, Dalian University of Technology, Dalian 116024, China; xiaoshayingdd@mail.dlut.edu.cn (P.Z.); xiaoyu1032@mail.dlut.edu.cn (X.X.); d.wang@dlut.edu.cn (D.W.); wangld@dlut.edu.cn (L.W.); 2Key Laboratory for Precision and Non-Traditional Machining Technology of Ministry of Education, Dalian University of Technology, Dalian 116085, China; yanying@dlut.edu.cn

**Keywords:** liquid dispensing, droplet, high-viscosity adhesive, critical contact force

## Abstract

The dispensing resolution of high-viscosity liquid is essential for adhesive micro-bonding. In comparison with the injection technique, the transfer printing method appears to be promising. Herein, an analytical model was developed to describe the dynamic mechanism of squeezing-and-deforming a viscous droplet between plates in a transfer printing process: as the distance between plates decreases, the main constituents of contact force between the droplet and substrate can be divided into three stages: surface tension force, surface tension force and viscous force, and viscous force. According to the above analysis, the transfer printing method was built up to dispense high-viscosity adhesives, which replaced the geometric parameters, utilized the critical contact force to monitor the adhesive droplet status, and served as the criterion to trigger the liquid-bridge stretching stage. With a home-made device and a simple needle-stamp, the minimum dispensed amount of 0.05 nL (93.93 Pa·s) was achieved. Moreover, both the volume and the contact area of adhesive droplet on the substrate were approximately linear to the critical contact force. The revealed mechanism and proposed method have great potential in micro-assembly and other applications of viscous microfluidics.

## 1. Introduction

Adhesive micro-bonding mainly refers to joining micro-components with liquid adhesive at approximately the picoliter to nanoliter level. In comparison with welding, thermal bonding, electrostatic bonding etc., it has many advantages including no requirement of high temperature and pressure, simple implementation, and low stress [1,2,3]. Therefore, it is widely used in assembly of microsystems such as micro-gyroscopes, micro-accelerometers etc. [4,5]. Since the bonding strength is generally proportional to the viscosity, a high-viscosity adhesive (>1 Pa·s) is often selected to ensure the joint strength. However, excessive adhesive at the joint will lead to creeping, contamination etc., whereas insufficient adhesive will directly lead to the decline of joint strength [6,7,8]. Obviously, the dispensing resolution is essential for adhesive micro-bonding. Traditionally, the injection has been often used for liquid dispensing [9,10]. However, to obtain the sub-nanoliter liquid, the inner diameter of a hollow needle has to be decreased to a few microns, thus the flow resistance caused by high-viscosity will become very high and the needle is easily blocked too [11]. Therefore, the application of injection is still difficult for the dispensing of high-viscosity adhesive in sub-nanoliter resolution.

Alternatively, the method of transfer printing appears to be promising, in which a stamp is used to dip liquid and then the dipped liquid is transferred onto the substrate by contact printing. Since it is based on surface flow rather than internal pipe flow, the effect of high flow resistance can be significantly reduced. Transfer printing with microarray has been utilized to produce micropatterns of various biomolecules and photoresists [12,13]. However, the viscosity in transfer printing was mostly limited to no more than 1 Pa·s [14]. Thus only effects of surface tension and solute-diffusion were considered in detail and viscous effect was often neglected. 

The essence of common transfer printing is a break-up process of the liquid-bridge between an upper plate (stamp) and a lower one (substrate). Previous research has indicated that the initial geometric parameters at the beginning of liquid-bridge stretching have critical influences on the final transferred volume on the substrate. For example, Zhang et al. studied the liquid-bridge break-up phenomenon between two cylindrical rods. It was found that the transfer ratio increases about from 70% to 90% with the increase of contact area of the droplet on the lower rod from 314 to 2826 mm^2^, following the decrease of distance between two rods [15]. Chen et al. investigated the liquid-bridge break-up in the microliter level. Their results indicated that the transfer ratio increases by about two times as the initial distance of the two plates decreases from 0.778 to 0.302 mm [16]. Huang et al. developed two-dimensional (2D) axisymmetric models about the stretching of the liquid-bridge and compared the theoretical predictions with the previously published experimental data (Chen et al.). The resulting predictions were consistent with the experimental observations. The prediction results also showed that the factor that mainly dominates the transfer ratio changes from the wettability difference between two plates to the viscous force with the increase of capillary numbers [17]. However, as the amount of transferred volume reduces to the sub-nanoliter level, the initial distance of the two plates and contact area of fluid on substrate will decrease to a few microns and dozens of square microns respectively. It is inconvenient and time-consuming to measure and adjust them with optical microscopy for industrial applications. 

Herein, an analytical model was developed to describe the dynamic mechanism of squeezing-and-deforming of a droplet between plates. It showed that in the case of the squeezing process, the contribution of the viscous effect to the contact force and contact area of the droplet on the substrate is dominant, following the decrease of distance between stamp and substrate. According to the above, the transfer printing method was built up to dispense the high-viscosity adhesive, which replaced the geometric parameters, utilized the contact force to monitor the droplet status, and served as the criterion to trigger the followed stretching of the liquid-bridge. With a home-made device and a needle-stamp, the minimum dispensed amount of 0.05 nL (93.93 Pa·s) was achieved. Moreover, both the volume and contact area were approximately linear to the critical contact force. The method has a great potential in micro-assembly and other applications of viscous microfluidics.

## 2. Principles and Modeling

### 2.1. Transfer Printing Principle

The proposed transfer printing process can be divided into three stages: dipping-up, squeezing, and stretching, as shown in Figure 1. 

In the dipping-up stage, a needle-stamp is kept at a constant distance from the viscous liquid level and an electric potential is applied between the liquid and the stamp. Thus, driven by Coulomb force, some liquid is stretched up to contact the stamp. Then the stamp is lifted up until the liquid-bridge breaks and thus a droplet is dipped onto the stamp. Here we only used the method to acquire the constant initial volume of droplet. An in-depth explanation of the electrokinetic phenomenon can be found in other references [18,19,20]. In the squeezing stage, the stamp with the initial droplet is positioned above the substrate. Then it slowly moves down to contact and then squeeze the substrate, during which the contact force *F*_C_ is generated by the squeezing effect on the substrate. Thus the *F*_C_ is defined as the force of the droplet acting on the substrate and measured with a force sensor. As a result, a liquid-bridge between the stamp and the substrate is formed and then deformed with continuous squeezing. Once the contact force *F*_C_ reaches a defined threshold (critical contact force *F*_CC_), the stretching stage is triggered and the stamp is lifted up until the liquid-bridge is pulled off. Consequently, part of the liquid is transferred to the substrate.

### 2.2. Modeling of Squeezing-and-Deforming of Initial Droplet

In the squeezing stage, the stamp keeps a constant descent velocity of *U* to squeeze the initial droplet. The contact force *F*_C_ on the substrate mainly comes from the contribution of the viscous force (*F*_v_) and surface tension force (*F*_s_). Thus, the starting point is the fundamental Navier–Stokes equation under cylindrical coordinates (*r*, *θ*, *z*) as shown in Equation (1):(1)$$\rho \frac{\partial \vec{u}}{\partial t}+\rho \vec{u}\cdot \nabla \vec{u}=\vec{f}-\nabla p+\mu \nabla^{2}\vec{u},$$
where *ρ* is the liquid density, *t* is the squeezing time, $\vec{u}$ is the flow velocity, $\vec{f}$ is the body force, *p* is the pressure, *µ* is the liquid viscosity. In order to solve Equation (1), some assumptions for the formed liquid-bridge between the needle-stamp and the substrate are made as follows:The liquid-bridge is axisymmetric and incompressible with low Reynolds number *Re* (Re<1, ∂p/∂θ=0, ρ∂u⇀/∂t+ρu⇀⋅∇u⇀≈0).The wettability of the stamp is similar to that of the substrate, thus the contact angle *α_i_* on the needle-stamp is approximately equal to the contact angle *β_i_* on the substrate, and the meniscus of the liquid-bridge is assumed to be the concave surface of the spherical cap, shown in Figure 2.The height of the liquid-bridge *h_i_* is much smaller than the contact radius *r_i_* between the liquid-bridge and the substrate. Then the slit between stamp and substrate is formed, and the pitch of slit is equal to *h_i_* (∂p/∂z=0) [21].The gravity in sub-nanoliter level can be ignored under low Bond number *Bo* ($Bo<1,\;\vec{f}=0$). Thus, the dominating terms are those considering the pressure gradients and the shear stress derivatives in the *z*-direction. Equation (1) can therefore be simplified to Equation (2).

The calculation of Reynolds and Bond numbers is shown in Section 4.1 in detail.

(2)$$\frac{\partial p}{\partial r}=\mu \frac{\partial^{2}u_{r}}{\partial z^{2}}.$$

Based on the boundary conditions *u_r_*
*=* 0 at *z* = 0 and *u_r_* = 0 at *z* = *h_i_* [22,23], the velocity distribution function can be solved by:(3)$$u_{r}=\frac{\partial p}{\partial r}\left(\frac{1}{2\mu }z^{2}-\frac{h_{i}}{2\mu }z\right).$$

This velocity distribution satisfies the mass conservation equation $-\pi r^{2}U=\int_{0}^{h_{i}}2\pi ru_{r}dz$, thus Equation (3) can be induced to:(4)$$p(r)=\frac{3\mu U}{h_{i}^{3}}r^{2}+p_{C}.$$

In the Equation (4), the boundary condition $p\left[r_{i}-\left(h_{i}/2\cos\alpha_{i}-h_{i}\tan\alpha_{i}/2\right)\right]$ on the meniscus surface is equal to $\Delta p_{L}$. $\Delta p_{L}=\gamma (1/R_{1}+1/R_{2})$ is the Laplace pressure, *γ* is surface tension, $R_{1}=r_{i}$ and $R_{2}=h_{i}/2\cos\alpha_{i}$ are the principal curvature radii of meniscus. Thus the pressure distribution function can be expressed as Equation (5) based on the boundary condition.

(5)$$p(r)=\frac{3\mu U}{h_{i}^{3}}\left\{r^{2}-{\left[r_{i}-\left(\frac{h_{i}}{2\cos\alpha_{i}}-\frac{h_{i}\tan\alpha_{i}}{2}\right)\right]}^{2}\right\}+\gamma \left(\frac{1}{r_{i}}+\frac{1}{h_{i}/2\cos\alpha_{i}}\right).$$

Then the contact force *F*_C_ on the substrate is given by
(6)$$\begin{array}{l} F_{C}=F_{v}+F_{s}=-\int_{0}^{r_{i}}p(r)2\pi rdr=\frac{-6\pi \mu U}{h_{i}^{3}}\left[\frac{r_{i}^{4}}{4}-\frac{r_{i}^{4}}{2}{\left[r_{i}-\left(\frac{h_{i}}{2\cos\alpha_{i}}-\frac{h_{i}\tan\alpha_{i}}{2}\right)\right]}^{2}\right]-\\ \left[\pi \gamma r_{i}^{2}\left(\frac{1}{r_{i}}+\frac{1}{h_{i}/2\cos\alpha_{i}}\right)\right] \end{array}$$
where the relationship between *h_i_* and *r_i_* can be obtained by the volume conservation before and after being squeezed: $\frac{2}{3}\pi h_{0}^{3}=\pi r_{i}^{2}h_{i}-\frac{\pi }{3}\left(\frac{h_{i}}{2\cos\alpha_{i}}-\frac{h_{i}\tan\alpha_{i}}{2}\right)\left[\frac{3h_{i}^{2}}{4}+{\left(\frac{h_{i}}{2\cos\alpha_{i}}-\frac{h_{i}\tan\alpha_{i}}{2}\right)}^{2}\right]$. Therefore, the contact force *F*_C_ can be solved by the geometric parameters including the stamp descent velocity *U*, adhesive viscosity *μ*, contact radius *r_i_*, the slit pitch *h_i_*, contact angle *α_i_*, and initial droplet height *h*_0_.

In summary, the essence of this model is to control the deformation of the liquid-bridge profile by the contact force, and thus control the amount of transferred liquid that will follow. Moreover, the liquid-bridge deformation model proposed by Amirfazli et al. has also attracted our attention. The principle of this model is to obtain the profile of the liquid-bridge based on the effect of contact angle hysteresis (CAH), thereby controlling the liquid transfer [24]. Compared to the model of Amirfazli et al., the difference of two models is that the method of controlling the profile of the liquid-bridge is different. What the two models have in common is that the initial distance between the two surfaces affects the deformation behavior of the liquid-bridge during the squeezing process.

## 3. Materials and Methods

### 3.1. Materials and Instruments

The selected liquid for experiment was an epoxy-based structural adhesive (EA 0151 Part A, Henkel Loctite, Dusseldorf, Germany). Its surface tension and contact angle on steel and silicon wafer were measured by a drop shape analyzer (DSA100, Kruss, Hamburg, Germany). The viscosity was measured with a viscosity meter (NDJ-8S, Shanghai Precision Instrument, Shanghai, China). The density was obtained by the weighing method with an electronic balance (JA1003, Tianjin Precision Instrument, Tianjin, China) and a measuring cylinder (5 mL). Measurement of all the above parameters was performed in the clean room (constant temperature 25 °C), and the measured results are shown in Table 1.

The experimental dispensing device was home-made as shown in Figure 3A, which consisted of three units: dipping-up unit, dispensing unit, and positioning unit. The dipping-up unit mainly included a stainless-steel needle-stamp (Figure 3B), a gold-plated silicon wafer, and some fixtures for the needle-stamp and gold-plated silicon wafer. The dispensing unit mainly included the needle-stamp with the initial adhesive droplet, silicon wafer, and force sensor (resolution 1 mN, LH-S09A-200g, Shanghai Sensing Technology, Shanghai, China). The silicon wafer was placed on the force sensor, shown in Figure 3C. The positioning unit was used to move the stamp in three dimensions. In addition, the adhesive volume *V* and its contact area *S*_a–s_ on the substrate could be measured with a 4× digital microscope. The measured transferred volumes were verified with a confocal laser scanning microscope (CLSM, VK-X200, Keyence, Osaka, Japan), as shown in Figure 4G. 

### 3.2. Experimental Process

The experimental process is shown in Figure 4A–F: (A) some adhesive was stretched up by Coulomb force under a suitable electric potential; (B) a liquid-bridge between the needle-stamp and the liquid level of adhesive was formed; (C) an adhesive droplet was dipped onto the needle-stamp with the liquid-bridge breakup; (D) the initial adhesive droplet was squeezed under the increasing contact force *F*_C_, during which a liquid-bridge was formed and deformed; (E) until *F*_C_ reached a defined value of 1 mN (*F*_CC_), the liquid-bridge was immediately stretched; (F) the liquid-bridge breaks and part of it was kept on the silicon wafer by the adhesive effect. The dispensing process was repeated five times for each defined contact force from 1 to 5 mN, shown in Figure 5.

### 3.3. Calculation of Adhesive Volume

The initial adhesive droplets were obtained with electrokinetic dipping-up as shown in the red dashed rectangle of partial enlarged drawing of Figure 4C. The transferred adhesive droplet is shown in a red dashed rectangle of Figure 4F. Their volumes *V* and contact areas *S*_a–s_ could be obtained by image characteristic extraction and calculation with Equation (7).
(7)$$V=\sum_{i=1}^{n}\pi x_{i}^{2}h_{p}\;(i=1,\;2,\;3\cdot \cdot \cdot \cdot \cdot \cdot );\;S_{a-s}=\pi x_{1}^{2},$$
where the initial and transferred droplets in the image can be divided into *n* micro-cylinders with the height of one unit pixel *hp* from top to bottom. The unit pixel *hp* is equivalent to 1.9 μm via spatial calibration. *x_i_* is the radius of the *i*-th micro-cylinder. The first micro-cylinder (*i* = 1) is in contact with the substrate, and so on. Similarly, the initial adhesive height *h*_0_ and the slit pitch *h_i_* could be obtained, shown in Table 2.

Moreover, the difference of transferred volumes that were respectively measured with optical imaging and CLSM was about one picoliter as shown in Figure 6A. Thus the measured method with optical imaging was considered to be accurate and reliable. Under the critical contact force *F*_CC_ from 1 to 5 mN, the variation of squeezed contact angles (Figure 4D) on the stamp and the substrate was obtained by image processing, shown in Figure 6B. Since the difference of squeezed contact angles *α_i_* and *β_i_* was less than 2°, *α_i_* could be considered to be approximately equal to *β_i_* in squeezing process.

## 4. Results and Discussions

### 4.1. Analysis of the Squeezing-and-Deforming Process of a Droplet

The main contribution to contact force *F*_C_ changed from surface tension force *F*_s_ to viscous force *F*_v_ with the decrease of slit pitch in the squeezing process. Herein, as the needle-stamp with initial droplet volume 0.15 nL pressed toward the substrate under *F*_C_ from 1 to 5 mN, the variation of *F*_v_ and *F*_s_ were summarized by MATLAB based on Equation 6 (other relevant parameters as shown in Table 1 and Table 2, Figure 6B). 

Generally, as a high-viscosity microdroplet was squeezed in the slit, both Re=ρU¯L/μ and $Bo=\rho gL^{2}/\gamma$ were very low due to the restriction of high-viscosity and volume of sub-nanoliter level, where *ρ* is the droplet density and *L* is the characteristic length (initial height *h*_0_ of the slit). $\bar{U}\approx S_{m}U/S_{c}$ is defined as the average velocity, which could be estimated by the volume flow conservation. *S_m_* is the average contact area at 1–5 mN, $S_{c}\approx 2r_{i}h_{i}$ is the longitudinal cross-sectional area of the slit (17.4–12.3 μm) at 1–5 mN. Thus $\bar{U}$ was 0.35–0.65 mm/min, *Re* was about 4.8 × 10^–6^ to 8.98 × 10^–9^, and *Bo* was about 0.9 × 10^–3^. Therefore, the squeezing-and-deforming process satisfied the requirements of low Reynolds and Bond numbers.

In the initial stage of squeezing, the initial droplet was squeezed under a smaller *F*_C_, the slit pitch decreased from 42.0 to 38.0 μm. At 42.0 μm, *F*_s_ was 7.3 × 10^–3^ mN, *F*_v_ was 6.1 × 10^–3^ mN, and *F*_s_ was greater than *F*_v_ (Figure 7A). As the slit pitch decreased, both *F*_s_ and *F*_v_ increased, but *F*_v_ increased faster, to 40.5 μm, *F*_v_ was equal to *F*_s_. As the slit pitch continued to decrease, *F*_v_ was greater than *F*_s_. When the slit pitch reduced to 17.4 μm with *F*_C_ of 1 mN, *F*_s_ was 2.4 × 10^–2^ mN, *F*_v_ was 0.9 mN (Figure 7B). The slit pitch decreased continuously to 12.3 μm with *F*_C_ of 5 mN, *F*_s_ was 6.3 × 10^–2^ mN, *F*_v_ was 4.8 mN (Figure 7C). Thus, *F*_v_ was far much greater than *F*_s_ as the slit pitch decreased from 17.4 to 12.3 μm.

Based on the above analysis, the main contribution to *F*_C_ could be divided into three stages in the squeezing process. In the initial stage, from the variation tendency of Figure 7A, *F*_C_ was primarily derived from the contribution of *F*_s_. As the slit pitch decreased to 40.5 μm, the contribution of *F*_s_ and *F*_v_ to *F*_C_ was equal. When the slit pitch continued to decrease from 17.4 to 12.3 μm (1–5 mN), *F*_s_ was increased about two times, but its contribution to *F*_C_ could be ignored compared to *F*_v_ (*F*_v_ >> *F*_s_). Thus, *F*_C_ mainly came from the contribution of *F*_v_ under this stage. Especially, the smaller slit pitch would make the dominant role of *F*_v_ even more pronounced.

As a result, when the high-viscosity adhesives (93.93 Pa·s) was squeezed from 1 to 5 mN (17.4–12.3 μm) in the paper, Equation (6) could be simplified to $F_{C}\approx \frac{-6\pi \mu U}{h_{i}^{3}}\left[\frac{r_{i}^{4}}{4}-\frac{r_{i}^{4}}{2}{\left(r_{i}-C\right)}^{2}\right]$, $C=h_{i}/2\cos\alpha_{i}-h_{i}\tan\alpha_{i}/2$, the volume conservation could be simplified to $\frac{2}{3}\pi h_{0}^{3}=\pi r_{i}^{2}h_{i}-\frac{\pi }{3}C\left(\frac{3h_{i}^{2}}{4}+C^{2}\right)$. According to the two simplified formulas, as the adhesive droplet was squeezed under 1–5 mN, the variation of contact area $S_{a-s}=\pi r_{i}^{2}$ in the theoretical calculation is shown in Figure 8. The results from theoretical calculation show that the *S*_a–s_ approximately linearly increases with the increasing of force, which is consistent with the experimental results.

In the two simplified formulas, since the value of *C* is a constant less than 3 μm under 1–5 mN (*r_i_ >> C*), Equation (6) can be further simplified to $F_{C}\approx 3\pi \mu Ur_{i}^{4}/2h_{i}^{3}$, and thus the volume conservation is $2\pi h_{0}^{3}/3\approx \pi r_{i}^{2}h_{i}$. By solving the above two equations, $F_{C}\approx 2\pi \mu Uh_{0}^{6}/3h_{i}^{5}\propto 1/h_{i}^{5}$ and $F_{C}\approx 81\pi \mu Ur_{i}^{10}/16h_{0}^{9}\propto r_{i}^{10}$ can be obtained. Thus, as the contact force *F*_C_ increases, the slit pitch (height of liquid-bridge) *h_i_* and contact radius *r_i_* are amplified in power. Consequently, as the force increases, a small change in the slit can lead to a slow increase in the contact area *S*_a–s_. This phenomenon can significantly lead to an improvement in the dispensing resolution. Interestingly, this analysis result further validated the linear increase in contact area *S*_a–s_ of Figure 8.

### 4.2. Discussion of Experimental Results

As shown in Figure 9, the experimental phenomenon that significantly improves the dispensing resolution based on force-feedback was observed. As the critical contact force *F*_CC_ increased from 1 to 5 mN, the contact area *S*_a–s_ increased almost linearly with the increase of *F*_CC_, shown in Figure 9A. Thus, it was easy to infer that the increasing of *S*_a–s_ would enhance the dragging ability caused by the adhesive effect, which caused more initial droplet to remain on the substrate. The research of Cai and Bhushan also showed that the contact area is the dominant factor to the strengthening of the adhesive effect [25,26]. Thus, the transferred volume also increased with *F*_CC_ increasing. As shown in Figure 9B, the approximate linear relationship of transferred volume with *F*_CC_ also proved the correctness of the above analysis. Furthermore, the increase in the transferred volume also meant a higher transfer ratio, shown in Figure 9C. 

The influence of initial adhesive volume on *S*_a–s_, transferred volume, and transferred ratio was also analyzed and the results are shown in Figure 9. Obviously, the increase of initial adhesive volume would result in the increasing of *S*_a–s_ (contact radius *r_i_*), which could be deduced according to the formula $F_{C}\approx 81\pi \mu Ur_{i}^{10}/16h_{0}^{9}$. According to this formula, since the viscosity *µ* and the descent speed *U* are constant, the contact radius *r_i_* monotonically increases with the increase of *h*_0_ (the larger initial volume) under the same *F*_CC_. Based on the above analysis, the increase of initial volume will lead to the increase of *S*_a–s_, so does the transferred volume and transfer ratio.

## 5. Conclusions

A theoretical model to describe the squeezing-and-deforming of highly viscous droplet was established and its dynamic mechanism was revealed. As the slit pitch decreased, the main constituents of contact force could be divided into three stages in turn: surface tension force, surface tension force and viscous force, and viscous force. Based on the analysis, the dispensing method of high-viscosity adhesives based on force-feedback was established with the home-made dispensing device. The results of experiment and theoretical calculation showed that when the critical contact force was in the range of 1–5 mN, the contact area of the adhesive droplet on the substrate was approximately linear to the critical contact force. Moreover, due to the effect of the contact area, the transferred volume in the experiment was approximately linear to the critical contact force too. As a result, the minimum dispensed amount of 0.05 nL (93.93 Pa·s) was achieved. The revealed mechanism and proposed method have great potential in micro-assembly and other applications of viscous microfluidics.

## Figures and Tables

**Figure 1 micromachines-10-00728-f001:**
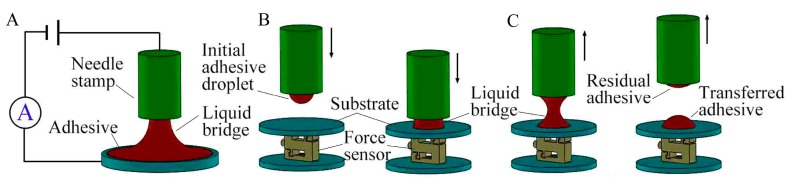
Adhesive dispensing with force-feedback: (**A**) dipping-up, (**B**) squeezing, and (**C**) stretching.

**Figure 2 micromachines-10-00728-f002:**
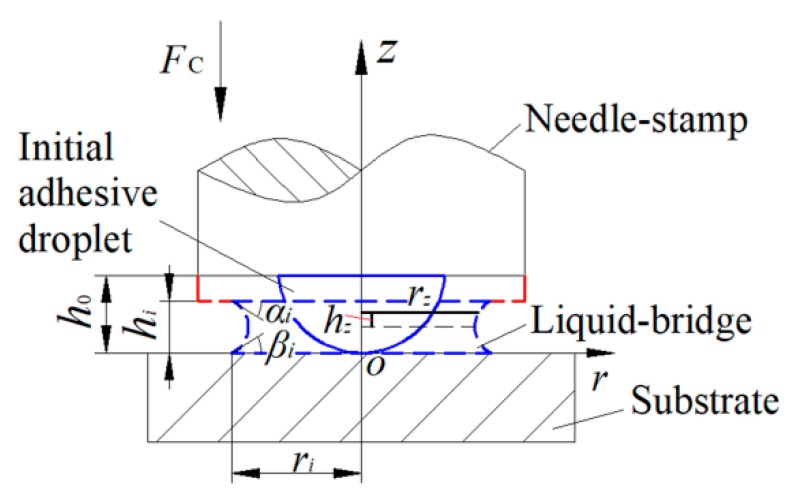
Schematic diagram of the initial droplet being squeezed.

**Figure 3 micromachines-10-00728-f003:**
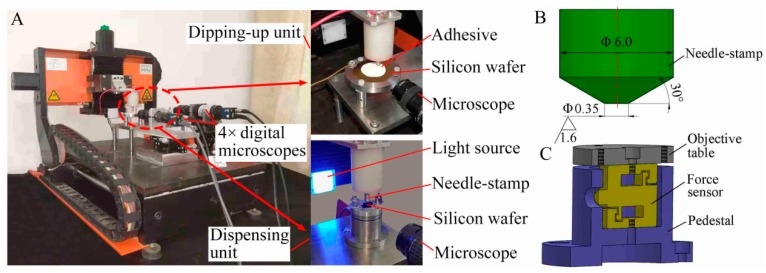
(**A**) Home-made experimental dispensing device with force-feedback, (**B**) needle-stamp, and (**C**) force sensor.

**Figure 4 micromachines-10-00728-f004:**
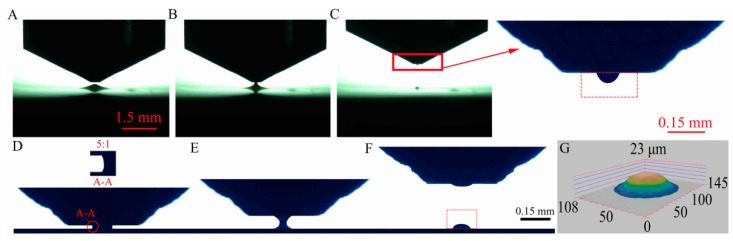
Adhesive dispensing with the needle-stamp in experiment under 1 mN: (**A**–**C**) dipping-up, (**D**) squeezing, (**E**,**F**) stretching, and (**G**) micrograph of transferred adhesive with a confocal laser scanning microscope (CLSM).

**Figure 5 micromachines-10-00728-f005:**
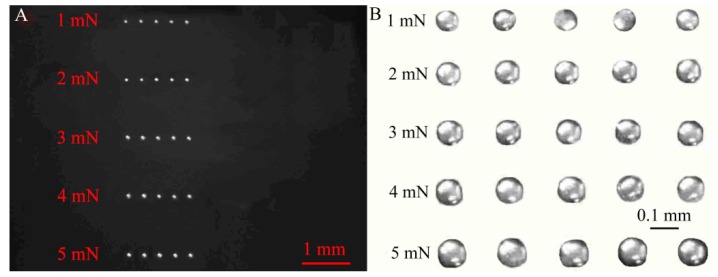
Adhesive droplets array before (**A**) and after (**B**) magnification under 1–5 mN.

**Figure 6 micromachines-10-00728-f006:**
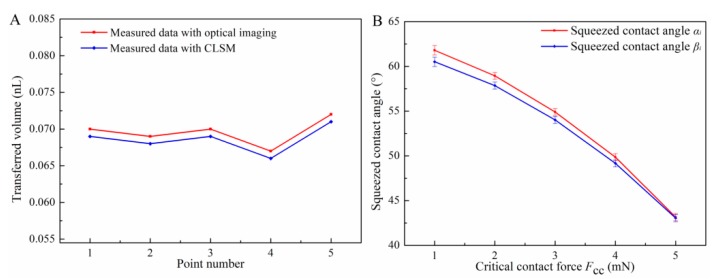
(**A**) Measurement difference of transferred volume at 1 mN, and (**B**) squeezed contact angles *α_i_* and *β_i_* at 1~5 mN with initial volume 0.15 nL.

**Figure 7 micromachines-10-00728-f007:**
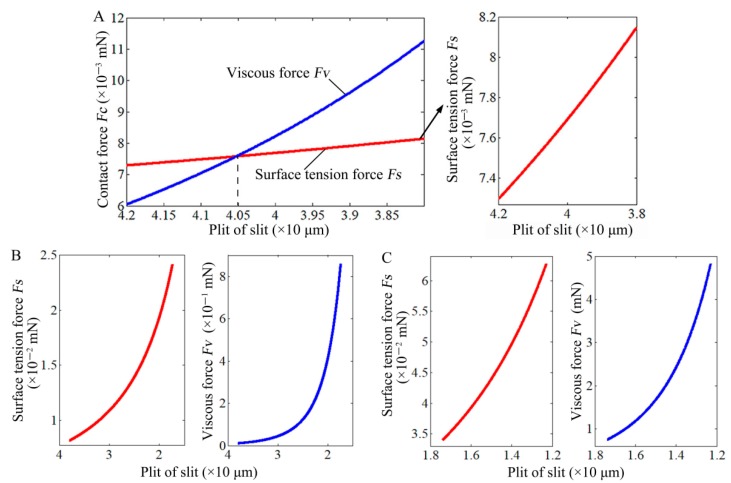
Variation of surface tension force (*F*_s_) and viscous force (*F*_v_) with the decrease of slit pitch at (**A**) initial squeezing, (**B**) 1 mN, and (**C**) 5 mN.

**Figure 8 micromachines-10-00728-f008:**
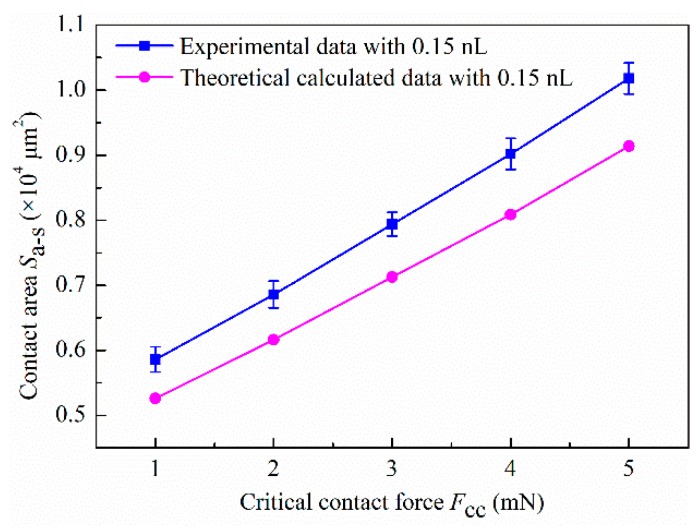
Variation of contact areas *S*_a–s_ of experiment and theoretical calculation with increase of critical contact force (*F*_CC_).

**Figure 9 micromachines-10-00728-f009:**
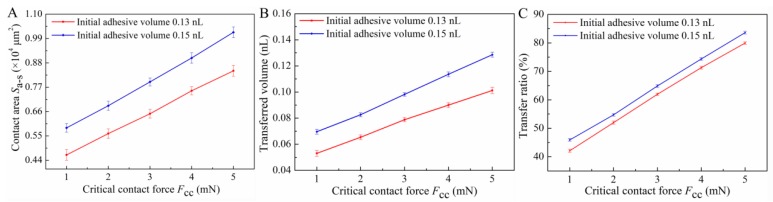
Variation of (**A**) contact areas *S*_a-s_, (**B**) transferred volume, and (**C**) transfer ratio with increase of *F*_CC_.

**Table 1 micromachines-10-00728-t001:** Properties of the adhesive.

Surface Tension *γ* (mN/m)	Density *ρ* (g/cm^3^)	Contact Angle *α*_0_ on Steel (°)	Contact Angle *β*_0_ on Silicon Wafer (°)	Viscosity *µ* (Pa·s)
36.96	1.84	74	71	93.93

**Table 2 micromachines-10-00728-t002:** Dispensing parameters.

Experiment Condition	Descent Velocity of Stamp *U* (mm/min)	Lifting Velocity of Stamp (mm/min)	Defined Contact Force *F*_CC_ (mN)	Initial Average Volume (nL)	Initial Average Height *h*_0_ (μm)	Pitch of Slit *h_i_* with 1–5 mN (μm)
I	0.09	60	1~5	0.13	39.0	15.9~11.3
II	0.09	60	1~5	0.15	42.0	17.4~12.3

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
