# Peer review of "Squeezing Dynamic Mechanism of High-Viscosity Droplet and its Application for Adhesive Dispensing in Sub-Nanoliter Resolution"

_micromachines, 2019, doi:10.3390/mi10110728_

Round 1
Reviewer 1 Report
Motivated by transfer-printing applications, the present paper reports
an analytical model for the deformation of a droplet
between two moving plates and the resulting forces (due to
viscosity and surface tension). Complementary experiments are also performed.
The paper is relatively easy to follow and the results will likely be
of interest from both fundamental and practical perspectives. I think
the paper should be published after the authors have made the following
revisions:
1. There is an additional paper on stretching liquid bridges that
should be referenced:
C.-H. Huang, M. S. Carvalho, and S. Kumar, Stretching Liquid Bridges with Moving
Contact Lines: Comparison of Model Predictions and Experiments, Soft Matter 12, 7457-
7469 (2016).
This paper provides theoretical predictions that are compared to the experiments of
Ref. 16 (Chen et al.) as well as other experiments.
2. On p. 7 the authors make reference to the work of Cai and Bhushan, but I did not see
these papers in the list of references. They should be added.
3. I believe that similar modeling and experiments have been carried out by
Amirfalzi et al. (references can be found in the paper of Huang et al. mentioned in point 1 above),
and perhaps also Bhushan et al. How does the authors' work differ from those previous works?
A discussion should be added to the paper
4. The paper doesn't really seem to compare model predictions to experimental observations.
Some discussion of this issue should be added to the paper, and a summary of the comparison
should be provided in the conclusions.
Reviewer 2 Report
My comments and suggestions are in the joint document.

Round 2
Reviewer 2 Report
The author have to clarify a bit the assumptions of the model (which are true, after my first reading). They have to speak of Reynolds number (Not low speed flow) and Bond number when neglecting gravity.
Then this paper will be ok for publication after a last strong English editing.
